# Effect of Portable, In-Hospital Extracorporeal Membrane Oxygenation on Clinical Outcomes

**DOI:** 10.3390/jcm11226802

**Published:** 2022-11-17

**Authors:** Anna L. Ciullo, Natalie Wall, Iosif Taleb, Antigone Koliopoulou, Kathleen Stoddard, Stavros G. Drakos, Fred G. Welt, Matthew Goodwin, Nate Van Dyk, Hiroshi Kagawa, Stephen H. McKellar, Craig H. Selzman, Joseph E. Tonna

**Affiliations:** 1Division of Cardiothoracic Surgery, Department of Surgery, University of Utah School of Medicine, Salt Lake City, UT 84132, USA; 2Department of Emergency Medicine, University of Utah School of Medicine, Salt Lake City, UT 84132, USA; 3Department of Surgery, Virginia Commonwealth University, Richmond, VA 23284, USA; 4Division of Cardiovascular Medicine, Department of Internal Medicine, University of Utah School of Medicine, Salt Lake City, UT 84132, USA; 5Division of Cardiothoracic Surgery, Evangelismos Hospital, Athens, AL 35611, USA; 6Division of Cardiothoracic Surgery, Department of Surgery, Intermountain Healthcare, Salt Lake City, UT 84132, USA

**Keywords:** cardiogenic shock, ECMO, ECPR

## Abstract

The time between onset of cardiogenic shock and initiation of mechanical circulatory support is inversely related to patient survival as delays in transporting patients to the operating room (OR) for venoarterial extracorporeal membrane oxygenation (VA ECMO) could prove fatal. A primed and portable VA ECMO system may allow faster initiation of ECMO in various hospital locations and subsequently improve outcomes for patients in cardiogenic shock. We reviewed our institutional experience with VA ECMO based on two time periods: beginning of our VA ECMO program and from initiation of our primed and portable in-hospital ECMO system. The primary endpoint was patient survival to discharge. A total of 137 patients were placed on VA ECMO during the study period; *n* = 66 (48%) before and *n* = 71 (52%) after program initiation. In the second era, the proportion of OR ECMO initiation decreased significantly (from 92% to 49%, *p* < 0.01) as more patients received ECMO in other hospital units, including the emergency department (*p* < 0.01) and during cardiac arrest (12% vs. 38%, *p* < 0.01). Survival to hospital discharge was equivalent between the two groups (30% vs. 42%, *p* = 0.1) despite more patients being placed on ECMO during ongoing cardiac arrest. Finally, we observed increased clinical volume since initiation of the in-hospital, portable ECMO system. Developing an in-hospital, primed and portable VA ECMO program resulted in increased clinical volume with equivalent patient survival despite a sicker cohort of patients. We conclude that more rapid deployment of VA ECMO may extend the treatment eligibility to more patients and improve patient outcomes.

## 1. Introduction

Despite improvements in management of cardiogenic shock, it remains associated with high morbidity and mortality [1]. The time between the onset of cardiogenic shock and initiation of circulatory support is inversely related to patient survival [2,3,4]. The use of venoarterial extracorporeal membranous oxygenation (VA ECMO) as a temporary means of cardiac support following shock has drastically increased over recent years. In comparison to conventional cardiopulmonary resuscitation (CPR), extracorporeal cardiopulmonary resuscitation (ECPR) has shown improved patient survival with regard to in-hospital cardiac arrest [5]. While ECMO has traditionally been implemented in the operating room setting, delays inherent to transporting a patient to the OR may present a barrier in obtaining rapid circulatory support and ultimately prove fatal. In light of this dilemma, our institution developed an in-hospital, primed and portable VA ECMO circuit that is able to be deployed in a variety of hospital locations starting in April of 2015 [6,7]. The multidisciplinary ECMO team is activated via paging protocol, with the aim of facilitating quicker time-to-initiation of circulatory support via percutaneous cannulation in the non-operating room setting [8].

Our study sought to compare patient outcomes before and after initiation of a portable VA ECMO system. We hypothesized that a primed and portable ECMO circuit would lead to increased patient survival.

## 2. Materials and Methods

Following Institutional Review Board approval, we performed a retrospective chart review of all patients placed on VA ECMO at our institution dating from November 2009–July 2017. Patients were classified into one of two cohorts based on whether ECMO was initiated prior to or after implementation of the portable ECMO program which began in April 2015. Exclusion criteria included patients who were placed on venovenous (VV) ECMO, as well as those who were placed on VA ECMO at an outside facility prior to being transferred to our institution. The primary end point was survival to hospital discharge. Secondary end points included associated morbidity, as well as the impact this program had on our institution’s clinical volume of VA ECMO patients.

Pre-determined data points were pulled from patient records and entered into our prospectively collected ECMO database. Exported values were then analyzed using either Pearson’s chi-squared tests for categorical variables, and independent sample t-tests for continuous variables. Clinical significance was defined as a *p*-value ≤ 0.05.

The descriptive statistics for continuous and categorical variables were performed using JMP (SAS Inc., Cary, NC, USA). This has been added to the manuscript.

## 3. Results

A total of 137 patients were placed on VA ECMO during our study period and were included in the study. Of those, 66 patients (48%) were placed on VA ECMO prior to program initiation, and 71 patients (52%) thereafter. A majority of these patients were male (69%), with a mean age of 55 years old at time of ECMO cannulation. Patient age ranged from 18 to 87 years old. Ethnicity was predominantly Caucasian (84%), and non-ischemic cardiomyopathy comprised 55% of heart failure etiology. We found no statistical significance between the two study groups with regard to patient demographics (Table 1). The second cohort, however, was significantly sicker as evidenced by the proportion of patients placed on ECMO during or following cardiac arrest in our post-program era and further evidenced by including patients receiving ECMO for out of hospital cardiac arrest (OHCA) (0% vs. 37%, *p* < 0.01). The proportion of patients placed on ECMO secondary to post-cardiotomy failure did not change between the two groups (34% vs. 32%).

In efforts to control for varying degrees of baseline acuity between the two groups, in addition to cardiac arrest, we looked at systolic blood pressure, creatinine and vasopressor requirement at time closest to ECMO implementation. Creatinine levels were found to be similar between the two groups (2.0 vs. 1.8, *p* = 0.21), as were vasopressor requirements (100% vs. 100%). However, mean systolic blood pressures (SBP) measured at the time of, or shortly after, ECMO initiation were found to be significantly lower in the latter cohort (89 mmHg vs. 78 mmHg, *p* < 0.01).

With regard to procedural data (Table 2), we noted a significant decrease in the number of patients placed on ECMO in the OR in the post-program era (92% to 49%, *p* < 0.01). While exact times from the onset of shock to initiation of ECMO were not available, initiation of ECMO in the latter cohort did not rely on immediate OR availability and avoided delays inherent in transportation. The proportion of patients cannulated peripherally did not change between the two groups (61% vs. 72%, *p* < 0.44), however we found a significant increase in the placement of distal perfusion cannulas for peripherally cannulated patients in the latter era (15% vs. 45%, *p* < 0.01). Of patients placed on peripheral ECMO support, our post-program era saw a decrease in arterial cannula size (21 ± 3.6 vs. 17 ± 3.1 French, *p* < 0.01). Total time on ECMO remained unchanged between the two groups. Patients cannulated in the first era required ECMO support for an average of 6.6 days, with those cannulated in the second era requiring an average of 6.3 days (*p* = 0.84).

Thirty-five patients (53%) in the pre-program era survived to ECMO decannulation, compared to 37 (52%) patients in the post-program era (*p* = 0.91). Patient survival to hospital discharge was equivalent between the two groups (30% vs. 42%, *p* = 0.15) despite sicker patients in the more recent cohort as evidenced by severity of shock and more patients undergoing ECMO initiation during cardiac arrest. When stratified by location of cardiac arrest, survival to discharge for patients with in hospital cardiac arrest (IHCA) qualitatively improve between our two cohorts (30% vs. 46%, *p* = 0.07). OHCA in the latter cohort was associated with 30% survival to hospital discharge.

In examining morbidity associated with VA ECMO (Table 3), we appreciated no statistical difference in cerebrovascular accident (CVA) occurrence between the two groups (7.6% vs. 13%, *p* = 0.32). Three patients were found to have evidence of anoxic brain injury in the first era, and 6 in the second era (4.5% vs. 8.5%, *p* = 0.50). Limb-related ischemia was found to be quantitatively higher in the earlier era compared to that of the latter (15.6% vs. 6.9%, *p* = 0.11). Three amputations were reported in the pre-program era, and 2 amputations reported in the post-program group (4.6% vs. 2.8%, *p* = 0.67). Renal function was measured via creatinine values, and there was found to be no difference between the two groups either at time of death (2.0 vs. 1.9, *p* = 0.81) or hospital discharge (1.4 vs. 1.6, *p* = 0.34).

We observed a significant increase in the clinical volume of patients requiring extracorporeal cardiac support. Our institution went from placing an average of 10 patients/year on ECMO support to 26 patients/year following program implementation (*p* < 0.01).

This section may be divided by subheadings. It should provide a concise and precise description of the experimental results, their interpretation, as well as the experimental conclusions that can be drawn.

**Table 1 jcm-11-06802-t001:** Patient demographics and ECMO data pre- and post-portable program initiation.

Variable	Before Program Initiation (*n* = 66)	After Program Initiation (*n* = 71)	Total (*n* = 137)	*p*
Age, years (mean, SD)	53 (16)	56 (13)	55 (15)	0.14
Gender (*n*, %)				0.65
Male	47 (71%)	48 (68%)	95 (69%)	
Ethnicity (*n*, %)				
White	58 (88%)	57 (80%)	115 (84%)	
Hispanic or Latino	4 (6.0%)	6 (8.5%)	10 (7.3%)	
Black or African American	1 (1.5%)	0 (0.0%)	1 (0.7%)	
Other	3 (4.5%)	8 (11%)	11 (8.0%)	
Etiology (*n*, %)				0.24
Non-ischemic	40 (61%)	36 (51%)	76 (55%)	
Cardiac arrest (*n*, %)	8 (12%)	27 (38%)	35 (26%)	<0.01
In-hospital	8 (100%)	17 (63%)	25 (71%)	
Out-of-hospital	0 (0.0%)	10 (37%)	10 (29%)	
Postcardiotomy failure	23 (34%)	23 (32%)	46 (34%)	
Creatinine at time nearest to ECMO placement	2.0	1.8	1.9	0.21
SBP at time closest to ECMO placement (mmHg)	89	78	83	0.01
Vasopressor requirement	66 (100%)	71 (100%)	137 (100%)	

ECMO, Extracorporeal Membrane Oxygenation; SBP, Systolic Blood Pressure.

**Table 2 jcm-11-06802-t002:** Procedural data.

Variable	Before Program Initiation (*n* = 66)	After Program Initiation (*n* = 71)	Total (*n* = 137)	*p*
Hospital location of cannulation (*n*, %)				<0.01
OR	61 (92%)	35 (49%)	96 (70%)	
Non-OR	5 (7.6%)	36 (51%)	41 (31%)	
Cannulation strategy (*n*, %)				0.44
Central	21 (32%)	20 (28%)	41 (30%)	
Peripheral	40 (61%)	51 (72%)	91 (66%)	
Placement of DPC (*n*, %)				0.01
Yes	7 (15%)	23 (45%)	30 (33%)	
No	30 (65%)	28 (55%)	51 (57%)	
Time on ECMO in days (mean, SD)	6.6 (6.2)	6.3 (6.3)	6.4 (6.3)	0.84
Cannula Size, French (mean, SD)				
Arterial	21 (3.6)	17 (3.1)	18 (3.8)	<0.01
Venous	25 (3.1)	24 (2.0)	24 (2.3)	0.38

DPC, Distal Perfusion Catheter; ECMO, Extracorporeal Membrane Oxygenation; OR, Operating Room.

**Table 3 jcm-11-06802-t003:** Associated Morbidity and Mortality.

Variable	Before Program Initiation (*n* = 66)	After Program Initiation (*n* = 71)	Total (*n* = 137)	*p*
Neurologic Outcomes				
CVA	5 (7.6%)	9 (13%)	14 (10%)	0.32
Ischemic	1 (1.5%)	4 (5.6%)	5 (3.6%)	
Hemorrhagic	3 (4.5%)	4 (5.6%)	7 (5.1%)	
Unknown	1 (1.5%)	1 (1.4%)	2 (1.4%)	
Anoxic Brain Injury	3 (4.5%)	6 (8.5%)	9 (6.6%)	
Herniation	1 (1.5%)	2 (2.8%)	3 (2.2%)	
Limb ischemia	10 (15%)	5 (7.0%)	15 (11%)	0.12
Ischemia requiring amputation	3 (4.6%)	2 (2.8%)	5 (3.7%)	0.67
Renal function				
Creatinine, day of discharge (SD)	1.4 (0.8)	1.6 (1.1)	1.5 (1.0)	0.34
Creatinine, day of death (SD)	2.0 (1.3)	1.9 (1.0)	2.0 (1.2)	0.81
Survival (*n*, %)				
To ECMO removal	35 (53%)	37 (52%)	72 (53%)	0.91
To hospital discharge	20 (30%)	30 (42%)	50 (36%)	0.15
IHCA	20 (30%)	28 (46%)		0.07
OHCA	0 (0%)	3 (30%)		

CVA, Cerebrovascular Accident; ECMO, Extracorporeal Membrane Oxygenation; IHCA, In-Hospital Cardiac Arrest; OHCA, Out-of-Hospital Cardiac Arrest.

## 4. Discussion

This study was performed due to increasing popularity of VA ECMO usage in non-operative settings following cardiogenic shock, and more recently secondary to cardiac arrest. We sought to determine whether using a portable system of bringing ECMO to the patient was associated with improved patient outcomes at our institution. The principal findings of our study are first, equivalent patient survival between the two groups despite sicker patients in the post-initiation cohort. Second, this program led to key practice changes with fewer patients receiving ECMO in the OR and a transition to using distal limb perfusion catheters and smaller arterial cannulae. Finally, having a portable VA ECMO circuit primed and ready was associated with increased clinical volume at our center.

While other programs have established ECMO teams with similar protocols, little data exists surrounding clinical outcomes [3,4,5,6,7,9,10,11,12,13]. In Japan, Komindr et al., examined clinical outcomes following establishment of a multidisciplinary ECMO team, which demonstrated no change between two patient cohorts with comparable Apache II scores [12]. Likewise, Shinar et al., demonstrated improved survival with implementation of ECPR by emergency physicians following OHCA when compared to control groups at non-academic centers [10]. Yannopoulos et al., examined patient survival following ECMO initiation in the cardiac catheterization suite for refractory ventricular fibrillation/pulseless ventricular tachycardia (VF/VT) during OHCA, and found a 50% survival to discharge with intact neurological function [14].

To ensure no obvious baseline confounders existed that may have disproportionately affected patient outcomes in support of improved mortality, we examined systolic blood pressure, creatinine, and vasopressor requirement at time of ECMO implementation. We found our post-program cohort to be much sicker based on the greater proportion of patients receiving CPR, and a lower SBP at time of ECMO initiation, demonstrating greater hemodynamic instability. We used this in conjunction with an increased incidence of cardiac arrest to conclude that our post-program era was sicker at time of ECMO implementation when compared to our pre-program era. Despite a sicker cohort of patients, however, we appreciated equivalent outcomes with regard to patient survival to hospital discharge (30% vs. 42%, *p* = 0.1), with numbers approaching significance when stratified by location of arrest with IHCA between our two eras (30% vs. 46%, *p* = 0.07). The survival to discharge rate for patients following IHCA in the more recent era (46%) is comparable to that of international registry data. Most recent analysis of the Extracorporeal Life Support Organization’s data registry reports adult survival rates to hospital discharge at 41% [15]. This number represents ECMO used for cardiac support, excluding that used during extracorporeal cardiopulmonary resuscitation (ECPR). Reported survival rates to hospital discharge following ECPR is 29% [15]. Similarly, our observed survival rate for ECPR secondary to OHCA is 30%, on par with the registry data.

This study suggests that a primed and portable in-hospital VA ECMO program may extend treatment eligibility to patients by allowing for more rapid deployment of cardiac support. This is evidenced by the significant increase in ECMO cannulation performed in various settings outside of the OR (7.6% vs. 51%, *p* < 0.01), including the emergency department (ED) for patients with OHCA. More specifically, we found that 66% of OR cannulations in the post-program era occurred secondary to post-cardiotomy failure, compared to just 38% of OR cannulations in the pre-program era. This finding demonstrates a decreased OR need in establishing rapid circulatory support for non-cardiopulmonary bypass-related etiologies. Additionally, our study also demonstrated a numeric decline in ischemic events in peripherally cannulated patients in the post-program era (15.6% vs. 6.9%, *p* = 0.11). This group saw a significant decrease in arterial cannula size (21 vs. 17 French, *p* < 0.01), as well as an increase in placement of distal perfusion catheters (15% vs. 45%, *p* = 0.01), likely related to decreased ischemic complications. Our post-program era saw a decreased incidence of limb amputation despite a similar rate of peripheral cannulation (4.6% vs. 2.8%, *p* = 0.67), reiterating the value of perfusion cannulae in prevention of limb ischemia.

Likewise, our institution saw a significant rise in the overall volume of patients placed on VA ECMO following program initiation in April 2015. Prior to establishment of a primed and portable circuit, an average of 10 patients per year were placed on VA ECMO. Following the deployment of our program, we saw that average increase to 26 patients per year (*p* < 0.01). It can be assumed that the increase in clinical volume is a byproduct of expanding patient eligibility by allowing for establishment of circulatory support in hospital departments outside of the OR.

Our study is subject to a series of limitations. First, our data was collected via retrospective review of a prospectively collected data base. Additionally, given our predominantly Caucasian male patient demographic, this study may not be generalizable to all populations. Finally, we are unable to retrospectively identify the time of onset or presentation of shock in this cohort so demonstrating a quantitative decrease in time to initiation of ECMO is inferred by avoiding limitations in OR availability and transport times to OR.

## 5. Conclusions

After development of an in-hospital, primed and portable VA ECMO program, our institution saw equivalent survival rates in a sicker patient population with no increase in associated morbidity. This program gave rise to key practice changes, with more patients being placed on circulatory support outside of the OR, including in the ED following OHCA, as well as adaptation of smaller arterial cannula size in the setting of increased distal perfusion catheter placement in peripherally cannulated patients. Finally, we saw an increase in the total number of patients being placed on VA ECMO each year. We conclude that this program has allowed for a more rapid deployment of VA ECMO, and has extended treatment eligibility to a wider array of patients.

## Data Availability

Data and code are available from the authors upon reasonable request.

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
