# Peer review of "Effect of Portable, In-Hospital Extracorporeal Membrane Oxygenation on Clinical Outcomes"

_jcm, 2022, doi:10.3390/jcm11226802_

Round 1
Reviewer 1 Report
Predominantly, I sincerely appreciate the effort of the authors to save the life of many people using described devices and their willingness to deal with their experience. I have only some minor additional comments about the study:
Content suggestions:
1. Can the authors provide and possibly make a comment about the influence of the use of described devices to the oxygen saturation dynamics ? Was the oxygen saturation at the time of the initial admission of the patient to the hospital indicating the further prognosis of the patient ?
2. I cannot see described Figure 1 and 2 – I suppose that they were planned to be added as the supplementary file, but I cannot assess them.
3. To be complete, can I kindly ask the authors about the statistical program used in the study ? Just for the formal point of this matter, please, reveal it in the manuscript.
From my point of view, the manuscript can be published after minor revision.
Author Response
Thank you for these insightful questions. We appreciate your input and have adjusted our manuscript accordingly. Please see below for additional context regarding each bulleted point the reviewer outlined.
- This is an excellent question with regards to oxygen saturation as a marker of prognosis and disease severity. In this study, however, we utilized alternative markers of malperfusion or distal organ dysfunction (i.e., lactate, renal function, vasopressor requirement). This was in part because several patients experienced cardiac arrest, thereby severely hypoxemic, and not necessarily an applicable marker for clinical outcomes in this setting.
- References to Figures 1 and 2 have been removed. Thank you for pointing this out.
- Thank you for this clarifying question. The descriptive statistics for continuous and categorical variables were performed using JMP (SAS Inc, Cary, NC). This has been added to the manuscript.
Reviewer 2 Report
The study is interesting and well presented. I have only minor suggestions:
- I would suggest that you report the rate of vascular complications. I see that limb ischemia is reported, however, that would be an important information.
- According to recent studies, ECMO may be the best mechanical support especially when combined with further devices as unloading strategies (please see PMID: 33677732). According to the same paper, however, there's high risk of bleeding with ECMO, as compared to different devices. I would suggest that you discuss the above mentioned points and, if possible, that you provide more data on bleeding outcomes registered i the two study arms.
- Pag 5, lines 125-127: I would erase.
Author Response
We appreciate this engaging dialog regarding vascular complications in ECMO patients. As you alluded to, there are an array of vascular complications in mechanically supported patients. In this study we looked to define the most clinically significant vascular complications (i.e., limb ischemia and hemorrhagic stroke) as opposed to granular transfusion data. Additionally, it should be noted that all patients in this study were decannulated in the operating room, therefore any necessary vascular reconstruction was performed at the same time.
Could this reviewer possibly clarify which lines they suggested to be removed? Page 5 does not have lines 125-127, only 149-201.